# Mental health related stigma in a primary care setting in Karnataka, rural India: Service user and carer perspectives

Sarah J. Parry[1], Gurucharan Bhaskar Mendon[2], Mirja Koschorke[3], Adarsha Alur Manjappa[4], Arya Thirumeni[2], Graham Thornicroft[5], Anish V. Cherian[2‡], Petra C. Gronholm[5,6‡]*

1 South London and Maudsley NHS Foundation Trust, London, United Kingdom, 2 Department of Psychiatric Social Work, National Institute of Mental Health and Neurosciences, Bengaluru, India, 3 Centre for Global Mental Health, Institute of Psychiatry, Psychology & Neuroscience, King's College London, London, United Kingdom, 4 Department of Health and Family Welfare, District Mental Health Program (DMHP), Ramanagara, Karnataka, India, 5 Centre for Global Mental Health and Centre for Implementation Science, Health Service and Population Research Department, Institute of Psychiatry, Psychology & Neuroscience, King's College London, London, United Kingdom, 6 Centre for Global Mental Health, Department of Population Health, London School of Hygiene & Tropical Medicine, London, United Kingdom

‡ Anish V Cherian and Petra C. Gronholm are joint senior authors.
* petra.gronholm@kcl.ac.uk

## Abstract

Mental health related stigma and discrimination are significant issues requiring urgent global action. There is a particular lack of research on stigma in low-and middle-income countries. This study explores the experiences of service users and their carers who access primary care for treatment of mental health conditions in rural Karnataka, India. Purposive sampling was used to conduct qualitative interviews of service users and their carers. This study highlights several barriers to access mental healthcare that individuals face on a system level. The results also bring to light pervasive experiences of stigmatising attitudes and discriminatory behaviour that participants with mental health conditions face from within their communities. Cohesion within both the family and the community was highlighted as a priority for participants, using a "what matters most" approach. This study brings insights to contribute to the development of locally relevant anti-stigma interventions, promoting access to care, and the human rights and quality of life for individuals with mental health conditions in a rural community in Karnataka, India.

## Introduction

Stigma and discrimination experienced by people with mental health conditions are often said to be worse than the condition itself and contribute to barriers in accessing healthcare, employment and education, and damages relationships [1–3]. The way

**Data availability statement:** Public deposition of the data underlying our findings is not possible due to ethical reasons, as consent was not sought from participants to having their data shared (even if anonymised) through methods like public repositories or as a supplement to the manuscript, and as such sharing data in this way would breach compliance with the protocols ap-proved by the relevant research ethics boards that have approved this research (King's College London PNM Research Ethics Subcommittee). Rather, participants have consent to their data being available for others through requests to the project team, i.e. the manuscript authors, or alternatively via contacting the King's College London Research Ethics Office (rec@kcl.ac.uk). Anonymised data will be made to other researchers via request, and no reasonable request will be denied.

**Funding:** GT, MK, and PCG were supported by the UK Medical Research Council (UKRI) for the Indigo Partnership (MR/R023697/1) award. GT has recently been supported by the National Institute for Health and Care Research (NIHR) Applied Research Collaboration South London (NIHR ARC South London) at King's College Hospital NHS Foundation Trust. The views expressed are those of the author(s) and not necessarily those of the NIHR or the Department of Health and Social Care. For the purpose of open access, the author has applied a Creative Commons Attribution (CC BY) licence (where permitted by UKRI, 'Open Government Licence' or 'Creative Commons Attribution No-derivatives (CC BY-ND) licence' may be stated instead) to any Author Accepted Manuscript version arising from this submission. AC received funding from an Intra mural grant from National Institute of Mental Health and Neuroscience, Bengaluru, India.

**Competing interests:** The authors have declared that no competing interests exist.

stigma and discrimination are experienced varies within different societies and cultures, according to what matters most in that particular context [3–5]. The experience of stigma negatively impacts upon recovery outcomes as well as on quality of life for people with mental health conditions and is a barrier to access and provision of care [1,6,7].

It is widely recognised that there is a stark treatment gap between the number of people with mental health conditions worldwide and those accessing treatment, particularly in low- and middle-income countries (LMIC) [8]. There is a significant discrepancy globally between the priority of physical health care and mental health care, with mental health care often taking a lower priority in terms of budget and policy [1,9]. The stigma associated with seeking care for mental health conditions can inadvertently reinforce this treatment gap by reducing the demand for care, and the lack of demand reduces the pressure to raise the priority of mental health care funding and to support a greater supply of services [1,9].

Stigma has been shown to be a factor in delayed help-seeking in mental health [10,11]. However, the way stigma affects individuals, communities and healthcare systems varies between societies and is less well understood in LMICs where less research has been conducted [1–3]. As well as a treatment gap for mental health, there is a significant research gap in LMIC [3,9,12], although the evidence base in LMICs is emerging [3,13]. The recent Lancet Commission on Ending Stigma and Discrimination in Mental Health highlighted the importance of research focused on the perspectives of service users and carers, especially in LMICs [1]. Exploring the lived experience of people with mental health conditions in LMICs is crucial to understand the complexities underlying the gap between the number of people requiring and accessing support for their mental health [1,14]. High quality qualitative research sheds light on the nature of the barriers and facilitators related to the accessibility, quality and effectiveness of mental healthcare [1].

India's population was estimated to be approximately 1.4 billion in 2023, equivalent to nearly a fifth of the total world population [15]. India is governed by a federal system and comprises of 28 states and 8 union territories with healthcare being governed by the union and state governments [16]. India has private and public healthcare providers, as well as non-governmental organisations providing healthcare, with the majority of private healthcare located in urban centres and rural areas mainly being served by the public system [17]. There is a three-tier system of public community care with the most local service being primary health centres (PHC) serving populations of 20–30,000, community health centres (CHC) serving 80–120,000 and tertiary health care [16]. Primary health centres may follow up individuals who have received diagnoses at a tertiary level by specialists and have long-term close therapeutic relationships with the communities they serve.

In 2017, nearly 200 million people in India were estimated to have a mental disorder – 1 in every 7 people [18]. From 1990–2017, the proportion of people with mental disorders contributing to overall disease burden in India doubled. Access to care varies from state to state and India's first national mental health policy was launched in 2014 prior to the 2017 revised Mental Health Act [19]. The treatment gap for common

mental disorders has been estimated to be approximately 80% [20]. India has approximately 0.75 psychiatrists per 100,000 people [21]. Mental health conditions are mainly diagnosed at a tertiary or district level by psychiatrists as well as via the District Mental Health Programme. Medical officers at primary health centres are trained to identify mental health conditions and provide follow up with service users including ongoing prescribed treatment.

Given this context, the aim of this study was to explore the experiences of stigma and discrimination of service users, and their carers, attending a primary health centre in rural Karnataka, India for mental health conditions. The results of this study contribute to understanding the barriers in accessing mental health services that individuals face in rural communities such as Karnataka in India. Such insights can inform the development of stigma reduction interventions and contribute to the enhancement of the rights and quality of life of individuals facing mental health conditions, and improve access to care in the future [1].

## Methods

### Study design

This study used data collected as part of the Indigo PRIMARY study [22], a multisite qualitative study of experiences of stigma relating to mental health conditions in primary care settings across Asia, Africa and Europe. Methodological details for this broader study have been reported previously [22]. The site considered for this study was in rural India, in Ramanagara district within Karnataka state, at PHCs associated with the District Mental Health Programme. These data have previously been analysed using framework analysis techniques, considering data across all research sites [22]. The purpose of this study is to carry out a richer, more in-depth exploration of these data, though an inductive analysis of the data collected with service-users and carers in the site in India specifically.

### Participants and procedure

Participants were recruited through purposive sampling by staff at PHCs who identified people receiving care for their mental health at the PHC and their carers. Recruitment procedures emphasised assessing potential participants' capacity, and the voluntary nature of study participation. Those who were willing to participate were included and service users who were currently symptomatic or cognitively impaired were excluded. Participation in the study was initiated only after written, informed consent had been obtained. The sample size was dictated by the data collected in the Indigo PRIMARY study, where a sample of around 10–15 participants per research site was assessed as sufficient to reach data saturation [23,24]. Saturation was considered achieved when the interviews were deemed to not provide new information or insights, indicating that additional data collection is unlikely to further the understanding of the research question.

The recruitment and data collection took place between 1st June 2018 and 31st July 2019. Interviews were carried out by members of the research team from National Institute for Mental Health and Neurosciences (NIMHANS), experienced in conducting mental health research and trained to recognise and respond to signs of potential distress during data collection. The interviews were conducted in the Kannada language, in private settings where participants could freely express their views.

Interviews lasted 60–90 minutes and a semi-structured approach was used, with an interview guide addressing topics related to experiences of stigma within healthcare and community settings for service users and carers. Questions used in the interviews are available as supplementary material (appendix 1). The interviews were translated and then transcribed in English by the same researcher. A further researcher, fluent in both English and Kannada crosschecked the transcript with audio recordings and reviewed the translation independently.

### Data analysis

The transcribed data were inputted to NVivo (R14.23.2) to facilitate data analysis. The data were examined through thematic analysis [25]. An inductive method of thematic analysis was used, allowing themes to emerge directly from the data

without being influenced by pre-existing theories, enabling a rich, data-driven understanding of participants' experiences. This process involved an initial familiarisation and immersion in the data through repeated reading of the transcripts, followed by generating a set of initial codes representing emerging broader patterns and descriptors of the raw data. These were then analysed and organised, through considering clustering between similar codes and connections between these broader groupings. Themes were identified from the codes, reflecting latent constructs characterising these emerging code clusters. These emerging themes were refined through an iterative process of re-reading transcripts, revisiting the codes and themes and their connections, and reviewing and refining the code and theme names for more accurate representation of the experiences and constructs they captured. The lead author (SJP) led this process, and the emerging insights and thematic structure was discussed in a series of online meetings between authors. The themes were refined to final key themes and a visual representation of the thematic map created as part of this process [25]. Analysis was carried out by author SJP in collaboration with three other authors (AC, PCG, GM), to identify the key themes describing the entire data set. Two authors (AC, GM) work within the study context, and two authors (SJP, PCG) work externally to the setting. This range of perspectives reduced bias in the analysis whilst maintaining important knowledge of the context. All key themes were developed and agreed on by all four authors over a series of online meetings.

Following practical guidance for achieving trustworthiness criteria in thematic analysis conduct [26], the analytical procedures described above also attest to how trustworthiness was established in this research. For example, the analytical process involved prolonged engagement with the data, retained field notes and transcripts, documenting the evolving coding process through notes shared amongst the research team, researcher triangulation during coding, audit trail of code generation, diagramming to make sense of code generation, vetting themes and subthemes among the research team, team consensus of themes, and providing a detailed description of the process of coding and analysis.

### Ethical approval

Ethics approval was obtained from the PNM Research Ethics Subcommittee, King's College, London: Reference Number (No.) RESCMR-17/18–4109; approval date 23 February 2017 and NIMHANS Institutional Review Board: Reference Number (No.) NIMHANS/IEC(BEH.SC.DIV.)7th MEETING/2017. Permission was granted from Directorate of Health Family Welfare Servicers, Government of Karnataka, India, No: DD/Mental Health/10/18–19 dated 26.04.2018.

### Inclusivity in global research

Additional information regarding the ethical, cultural, and scientific considerations specific to inclusivity in global research is included in the supplementary materials (appendix 2).

### Results

A total of 14 qualitative interviews were carried out with nine service users and five carers. Demographic data was collected for the participants, detailed in Table 1.

The aim of this study was to explore the experiences of stigma and discrimination of service users and carers attending a primary health centre in rural Karnataka, India for mental health conditions. The results reflected four interconnected key themes relating to this research question: 1) Complex System, 2) Knowledge, 3) Stigma and Discrimination and 4) Social cohesion within family and community. Each theme also includes a set of subthemes. These themes and their interconnections are illustrated in Fig 1, and described in further detail below.

There is some overlap and connection between the four themes (see Fig 1). The themes "Knowledge" and "Stigma and Discrimination" feed into the theme "Social cohesion" and these three themes sit within the wider context of the theme "Complex System". Furthermore, themes 2 (Knowledge) and 3 (Stigma and Discrimination) interact with each other and there is some overlap and connection between these two themes. For example, the "Stigma and Discrimination" subtheme 3b (Myths about mental illness) does to some extent reflect aspects of the "Knowledge" theme 2. However, the

**Table 1. Demographic details of participants: service users (n = 9) and carers (n = 5).**

|  | Service Users | Carers |
|---|---|---|
| **Gender** | | |
| Male | 3 | 3 |
| Female | 6 | 2 |
| **Age (years)*** | | |
| <30 | 0 | 0 |
| 31–40 | 3 | 1 |
| 41–50 | 1 | 1 |
| 51–60 | 1 | 2 |
| 61–70 | 2 | 1 |
| >70 | 1 | 0 |
| **Marital Status** | | |
| Married | 9 | 5 |
| Single | 0 | 0 |
| **Religion** | | |
| Hindu | 9 | 5 |
| Other | 0 | 0 |

* one participant did not provide details of age.

data in theme 3 (stigma and discrimination) was more specifically related to stigma and discrimination than the data presented in theme 2.

## Complex systems

This theme captures the difficulties highlighted by participants in navigating accessing care for their mental health conditions. Participants described seeking help and support for their mental health through a variety of sources and via a number of different routes without clear referral pathways, sometimes simultaneously and sometimes sequentially (subtheme 1a). Participants described barriers at each stage of accessing care and continuing to seek support through the PHC where this study was carried out (subtheme 1b). There was not a single, uniform pathway leading to participants accessing help for their health condition, and several systemic barriers described in accessing consistent healthcare (subtheme 1b). This theme addresses the practical and logistical barriers that have arisen due to the complex system, rather than barriers in accessing care relating to knowledge and stigma; these are described in themes 2 "Knowledge" and 3 "Stigma and Discrimination", which sit within the context of this theme.

**Pathways to care.** Participants described frequently seeking access to help and support for mental health conditions through a variety of different sources, including healthcare at the hospital level prior to primary care or alternative sources such as through spiritual care. One caregiver stated:

*"I went to different private practitioners in Ramanagara. When it didn't cure kept on seeing different doctors."* (SU1)

The experience of seeking care from multiple sources before the primary health centre without one clear linear referral pathway was frequently mentioned, for example:

*"We took her to private hospital and district hospital for treatment initially. We did not know that medicines are available in PHC."* (CG2)

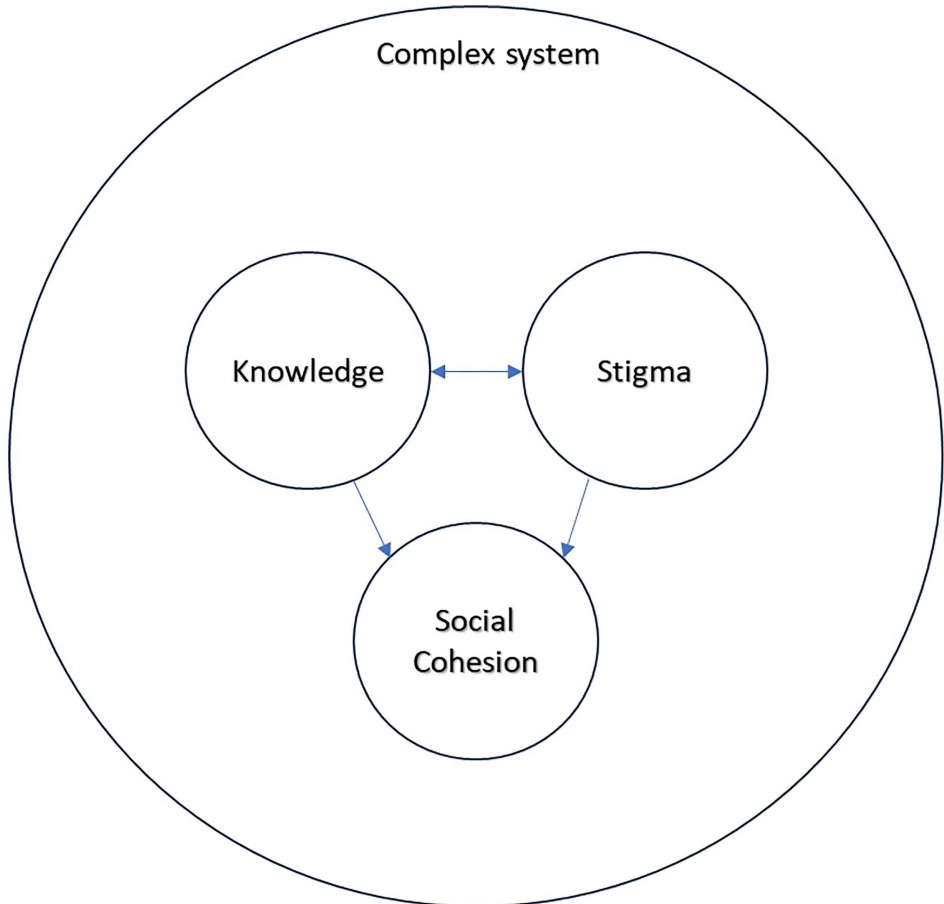

**Fig 1. Diagram of four key themes and their interconnections.**

Seeking support from spiritual sources was frequently mentioned either prior to seeking care from health services, or alongside healthcare, as illustrated in the following data extracts:

*"We went to temples and did poojas\*, but had no improvement. Then our neighbours told about taking treatment from Ramanagara District Hospital."* (CG3) (\*Pooja is a word used to describe a ritual of respect or worship to a god, e.g., visiting a temple, making an offering, chanting prayers).

*"I used to do poojas and prayers for her improvement. But there was no change… Later, I took her to District Hospital to get treatment."* (CG3)

Some participants continued to seek spiritual support alongside care at the primary health centre*:*

*"I did Poojas. There was some relief. Now also pooja gives me some satisfaction."* (SU3)

The variety of responses highlighted the heterogeneity in pathways for seeking care for mental health conditions and the lack of awareness of the possibility of attending the primary health centre for mental healthcare referral and follow up, despite it being the usual first point of access for other health issues.

**System related barriers to accessing care.** Several difficulties with accessing care at the primary health centre level and navigating the system were mentioned by participants including financial barriers, travel difficulties, medicine availability, human resources and long wait times. Many participants highlighted the lack of public transport to the primary health centre and that they must walk or use an autorickshaw, which was described as expensive, adding to the financial burden of seeking healthcare and the difficulty accessing the system:

*"We live in village, which is 3 kms far from PHC. It is difficult to reach PHC because there are no public transport facilities."* (CG5)

*"I do not have money to go to far places. I have to come by auto even to the PHC. I have to bear all expenses."* (CG3)

As well as transportation being a barrier to accessing care, the limitations of resources at the primary health centre was highlighted by participants in terms of human resources and medication supplies:

*"If there are two or three doctors at PHC and if they are specialized, there will be better services at the primary care centres."* (CG1)

*"Staffs should be regularly available to offer services to the patients. Especially if doctors are available every day, people will consider and accept care at PHC."* (CG1)

*"I have not been getting two medicines. I came twice and the medicine is unavailable. Those are two medicines that makes my daughter normal."* (CG3)

These barriers described the difficulties service users experienced in accessing the public healthcare system to get support for their mental health. As well as the complex journey to reach the system and the practical barriers to accessing the system participants experienced, they also experienced difficulty once within the system in the way of several limitations creating obstacles to accessing continued support for their mental health. Stigma-related influences constituted another barrier to accessing care, which is discussed in theme 3.

### Knowledge about mental health

Reflections around how much knowledge service users, their carers and health professionals had around the service user's mental health was a repeated and prominent theme in the data. The concept of "not knowing" in subtheme 2a about one's mental health (e.g., the name of your diagnosis, the name of your treatment, where/how to get further support) was a repeated concept. It is important to highlight that this was not assumed to be negative and participants lack of knowledge was not presented in an overtly negative or positive light (subtheme 2a). It was also noted that the way participants conceptualised their conditions was frequently with relation to physical symptoms and beneficial treatments were often biological (e.g., medication, behavioural activation) as described in subthemes 2b and 2c. Participants also expressed a desire for improving and increasing knowledge for carers and staff (subtheme 2d).

**Lack of knowledge.** Several participants highlighted that they do not have information about their diagnosis:

*"Doctors do not give information about illness. They just give medicines and ask them to take regularly and tells that the problem will be reduced on taking treatment."* (CG2)

However, it was not necessarily always the case that participants wanted to request this information from doctors:

*"I do not know the name of problem. I did not ask doctor and doctor also did not tell."* (CG2)

Half of the participants specifically said they did not know the name of their illness, answering simply *"I don't know."* when asked. Many also said they did not know the cause of their illness, again answering *"I don't know."*

Regarding knowledge about sources of support, when asked if they are treated differently or have a complaint what to do, half of the participants said that they "did not know" what to do.

Although half of participants said they would not know what to do if they had a complaint, some participants were aware of routes to raising concerns or complaints. However only one participant (SU3) stated *"I will complain to District Health Office".* Four others said more generally they would tell staff at the hospital. Two participants said they would not make a complaint, one stating:

> *"If I do not like treatment or any services or have any other problems, I will go to other place for treatment. I will not go for any complaints."* (CG5)

**Knowledge relating to diagnosis.** When asked "What do you call these problems you are getting treatment for?", only one participant was aware of their diagnosis (SU6): *"My problem is called fits."* Another participant said (SU4): *"I have some problem of brain."*

Of the fourteen participants, half did not know their diagnosis, one did know their diagnosis, and the remaining participants described their problems in terms of the symptoms, primarily biological or physical issues:

> *"My problems are that [an] angiogram has been done, I have blood pressure, sugar, abdominal pain, leg pain, body pain, sleeplessness, tension, difficulty in breathing."* (SU1)

> *"I was suffering from problems of headache, sweating, sleeplessness, pass stool for excessive times."* (SU2)

> *"I have chest pain, leg pain and eye pain."* (SU5)

Tension was a word frequently used to describe symptoms:

> *"She has too much tension."* (CG5)

> *"Tension is my problem."* (SU3)

**Knowledge relating to treatment.** Participants spoke about the positive benefits of support they have received for their health problems including medication, and alternative therapies. Beneficial treatments included medication, alternative/spiritual support and behavioural activation, rather than reference to psychological therapies.

> *"She is taking medicines from this PHC. She has also taken help from Ayurveda* [system of traditional medicine in India]. *After taking that, there is some improvement."* (CG5)

> *"We get medicines and there is some improvement."* (CG5)

Some participants were actively engaged in maintaining their good health:

> *"I have received training classes for 1 hour from this PHC on how to lead life, how to sleep well, I have been instructed to do exercise, walking and I do 5 kms walking."* (SU3)

When asked whether there are any human rights or advocacy organisations that support people with mental health conditions and their families, all participants said no, or that they did not know of any which is in keeping with the lack of such organisations in this area.

**Desire for more knowledge.** The importance of specialised mental health training and education to increase knowledge was highlighted by several participants, for both health professionals and for lay people:

> "*If we [carer] get information on how to take care of mental illness, it will be really helpful.*" (CG5)

> "*Staffs should know about mental illness and how to take care of patients.*" (CG2)

> "*PHC staffs are all good and helpful, but I need to tell my problems with a psychiatrist who can only actually help me. We need specialist help and services.*" (SU3)

### Stigma and discrimination

Experiences of stigma and discrimination were described by all the participants. Participants described their observations of negative attitudes from others related to their mental health condition (subtheme 3a) as well as incorrect and stigmatising beliefs around mental health conditions about their treatability or consequences (subtheme 3b). The largest subtheme of this theme was the experiences of negative social and behavioural consequences (discrimination) and being treated differently as a result of their mental health condition (subtheme 3c).

**Stigmatising attitudes.** Participants described observing negative attitudes from others regarding their mental health condition and the impact this has on them:

> "*People talk about my illness in a wrong way. How can I control them? I cannot do anything.*" (SU6)

> "*Other people think it as a different illness, not like fever or cough.*" (CG2)

The impact of negative attitudes from others was striking and one participant described the subsequent social isolation that resulted:

> "*They see us as strange persons. We used to not go out of our home because of this.*" (CG2)

**Myths about mental illness.** Some myths about mental health conditions were described by participants, contributing to the stigmatising attitudes described above:

> "*People think that the illness will not cure.*" (CG2)

> "*Sugar can be controlled on taking medicines, but mental illness cannot be controlled.*" (CG3)

> "*People think that mental illness cannot be treated.*" (SU2)

A specific myth about not being able to have children was described:

> "*People talk among themselves that I am having some mental problem, tell that I take medicines and I do not get child.*" (SU6)

**Discrimination.** Participants described the subsequent discrimination and ways they are treated by others as a result of their mental health condition, for example being treated differently:

> "*People behave differently. Some people won't show interest to talk to us while some others fight with us.*" (SU2)

> "*People look at persons with mental illness in a different way.*" (CG3)

A lack of respect was frequently mentioned

"*My family members do not listen to me and scolds me. Nobody respects me, especially my daughter in law and grand-children.*" (SU5)

"*Mostly people in my village treat me with less respect.*" (SU3)

There was a significant number of mentions of violence or physical mistreatment:

"*I have experienced violence from my husband.*" (SU4)

The social exclusion and rejection from others in light of their mental health condition was frequently mentioned:

"*People will remain far from such people who have this illness.*" (CG5)

Use of stigmatising language was described:

"*I had a fight with staffs there one day because they told that I am 'Brain Loose'.*" (SU9)

A lack of help from others was frequently described: "*No one else is helping.*" (CG3)

### Social cohesion within family and community

The most distressing stigmatising experiences of discrimination presented by participants were most frequently in the context of exclusion from the community and a prominent theme throughout the interviews was of service users and carers describing their experiences generally in the terms of how it related to their family or caregivers. The involvement of family in supporting with accessing and providing care was frequently described (subtheme 4a) and the subsequent impact on families and carers of the individuals experiences was also frequently described (subtheme 4b). Within this, the initial presentation to services was often described as being due to a breakdown of relationship or conflict within the family or community, highlighting the importance of social cohesion. Rather than seeking care as a result of symptoms, it was more commonly seen that families sought care when disruption to the family cohesion was experienced. Within theme 4 is also data relating to the positive experiences participants described relating to social cohesion (subtheme 4c).

**Importance of family.** The involvement of family in supporting participants to access care and also take part in activities of daily life was frequently described by participants:

"*My family members- wife and son- took me to NIMHANS.*" (SU8)

"*My daughter does not perform any activities of self-care such as eating, toileting, bathing, dressing on her own. I am doing everything for her.*" (CG3)

The supportive role of family was described for example when participants experience discriminatory behaviour or stigmatising attitudes:

"*My wife also does not listen to them* [those in the community expressing stigmatising attitudes and behaviours] *and she supports me.*" (SU6)

Furthermore, the impact of service user's health conditions on family members was described and participants spoke about their concerns and anxieties for the wellbeing of their family members experiencing difficulties with their mental health:

*"I was worried that she became like this. I was taking care of her."* (CG2)

*"I need my daughter to be well, I wanted to know if she will also become like other children and it would be good if staff can help me to know."* (SU2)

**Psychosocial impact of mental health condition.** Related to the importance of family, participants often described the nature of their problem with relation to the impact it had or was having on their family and community relationships. Often, the reason or initial presentation to services involved a difficulty within the family or community context:

*"I am getting tensed when others are arguing with me. It was started when we were building our new house. There were some family issues. Then I got all worries."* (SU9)

It was striking to note that anger or a family conflict was frequently cited as the first reason help was sought or the main way the mental health condition presented:

*"My mother was okay before two years and anger increased suddenly."* (CG1)

*"My son fights sometimes and we have worries when he behaves like that thinking why he is behaving like that with us. He does not listen to what we say or give respect to us."* (CG4)

*"She fights with me and scold in the evening when I come home after work without any reason."* (CG3)

Difficulties in the work place context were also a trigger to mental health deterioration as described by one participant:

*"I had work place related issues. The colleagues were cheating and taking money. Then, I got lot of pressure and stress and then I felt that my brain has got some problem."* (SU4)

**Describing positive experiences with healthcare staff.** Many participants described good relationships with healthcare staff:

*"All staffs and doctor are very cooperative at PHC."* (SU3)

*"Doctor in this PHC is really helping minded. She takes care well. Other staffs also understand my problems and support me."* (SU9)

Contrasting with the negative experiences of discrimination reported by several participants, there were also several comments about not being treated differently and being accepted by communities:

*"No one in our village behave in a different way. I did not experience like that. People ask us to do some Poojas and offer prayer in temples to have improvement."* (SU7)

*"No one has treated him differently till now. No one has caused any trouble to us."* (CG4)

## Discussion

The aim of this study was to gain an insight into the experiences of stigma and discrimination that service users and their carers attending a primary health centre in rural Karnataka, India, face relating to their mental health care. It is important to understand the stigma-related barriers that individuals in specific contexts face in accessing mental health services to inform the development of locally relevant anti-stigma interventions and improve access to care for people with mental

health conditions. In addition, such insights aim to improve quality of life and promote the rights of individuals with mental health conditions. Such studies are particularly needed in LMICs, where research to date is limited and where there are significant treatment gaps for mental health [1,12].

This study highlights barriers to accessing care on both a system level and also relating a lack of knowledge about mental health. This study brings to light the pervasive experiences of discriminatory behaviour due to their mental health condition that individuals in a rural community in India face, as is the case in other countries in the region [7,12,27]. Stigmatising attitudes (prejudice) and behaviours (discrimination) were described by participants from within their local communities and family contexts. It was noted that cohesion within the family and community was a priority for participants, and may be considered "what matters most" [4] for the participants in this study. The concept of "what matters most" is considered a helpful framework for understanding the contextual nuance and cultural influences on stigma [4,28]. Social cohesion and connection appears to be a culturally significant aspect of daily life in this study, that is compromised by the threat or experience of stigma and discrimination.

In India, as in most LMICs, many people with severe mental health conditions may be made homeless, or depend on their families financially and for support with personal and health care [29]. Families are therefore often closely involved with seeking healthcare and taking on roles that may be filled by health or social care services in other settings [29]. India is traditionally viewed as a collectivist society, where values such as social cohesion and community are held as a priority and where interdependence is promoted [30]. The value of social cohesion may be even higher in rural settings where cultural values may be more traditional, and studies have shown differences between experiences of discrimination of people with mental health conditions such as schizophrenia in India in rural compared to urban areas [31]. The finding of social cohesion and the importance of family being a key theme in this data is therefore in keeping with what is expected from existing literature and knowledge [7,30].

Rather than operating as individuals as might be seen in a Western context, participants in this study routinely described their experiences collectively, in relation to their family and community context, whether directly or indirectly [30,32]. The importance of cohesion and maintaining relationships within their family or community context was repeatedly highlighted as a priority for participants [32]. An interesting phenomenon noted by the authors was that many participants described positive experiences of their engagement with the primary health centre despite the barriers they also described. One possible explanation is that the importance of maintaining cohesion and good relationships with the professionals who care for them may have influenced participant answers in light of professionals having an ongoing role in their care [33]. The value of social cohesion above a more Western value of autonomy therefore may extend beyond family and society and influence experiences within healthcare settings [34].

Participants frequently described mental health conditions as presenting initially with a behavioural change causing conflict within the family or some form of dysfunction with relation to participation in society. Rather than a description of depression involving the cognitive symptoms (e.g., low mood, anhedonia, poor motivation, hopelessness, suicidality) more frequently, behavioural and somatic symptoms were described (poor sleep, poor appetite, irritability, conflict) and their impact on their function within the family or social context [35,36]. Interestingly, no participants described any of their symptoms or illnesses using language relating to thoughts or emotions (e.g., low mood, hopelessness). Furthermore, participants who described helpful aspects of treatment and care referred to biological tools such as medication and a healthy lifestyle.

Previous studies of the experience of stigma and discrimination of people with schizophrenia in India have highlighted that more internalised forms of stigma for example a sense of alienation is a more common experience than negative discrimination [37]. In this study, participants described the negative consequences of stigmatising attitudes from others in terms relating to their social isolation and exclusion from community, for example leading to them not leaving the home [38].

Participants frequently did not know their diagnosis or treatment plan, in contrast with the focus on a medical model of mental illness, patient knowledge, autonomy and rights that are often prioritised in a more Western context [39]. The lack

of knowledge was observed to be distinct from the experiences of stigma and discrimination that was described by participants, and was therefore separated into a different theme.

### Future recommendations

There are several implications of this study relating to clinical practice and future research opportunities. This study highlights the experiences of stigma and discrimination that individuals within a rural context in India are experiencing and the need for the development of relevant anti-stigma interventions which take into consideration the importance of social cohesion and also the practical and systemic barriers that are present for individuals to access care. Addressing stigma and discrimination are urgent global issues, which require immediate action [1]. Given the size of India and the diversity of its population, it would be beneficial for this research to be replicated with a wider range of participants in a variety of geographical areas across the country. In order to address the potential barrier that participants may face regarding speaking openly about their experiences of healthcare, and given the importance of social cohesion and hierarchical relationship with professionals, it would be beneficial to involve independent researchers separate from healthcare settings in carrying out further studies.

### Limitations

Certain limitations of this study are important to highlight. As interviews were carried out in Kannada and translated to English before being transcribed, the analysis is based on the translator's interpretation of the participants responses, rather than the direct responses themselves. However, the translators were confident in both languages and all translations were subsequently cross checked. We also acknowledge the limitation that for feasibility reasons, the analysis did not include independent coding by multiple authors or procedures to establish intercoder reliability. However, the lead author collaborated with other authors during the analysis process, ensuring coding was not biased by a subjective perspective. The insights gathered from the data remain valuable as despite the limitations, important rich data have been collected relating to the experiences of individuals with mental health conditions in rural India.

The participants of this study shared several similar characteristics; they were all married, all Hindu and from a rural district in India. Therefore the results may not necessarily be able to be generalised. However, qualitative research findings are not intended to be broadly generalisable, but rather intend to explore the experiences and insights of the specific participant group involved in the study. Furthermore, the shared characteristics of the sample strengthens the themes identified in these data as there were many similarities in the participants' context.

### Conclusion

The results from this study give a voice to individuals experiencing stigma and discrimination relating to their mental health, and their carers, providing a unique window into the experience of service users and carers from a rural location in India. Service users from LMICs, particularly in rural settings, are a group which are often overlooked, especially in research studies. The experiences of stigma and discrimination the service users and carers who participated in this study is vitally important to be recognised. There is a richness to these data that adds greatly to the body of research exploring service-user experience of stigma and discrimination, which is primarily focused on voices from high-income settings.

### Supporting information

**S1 File. Appendix 1. INDIGO-PRIMARY Topic guide: Interviews with service users (SU) and their family members.** (PDF)

**S2 File. Appendix 2. PLOS ONE 'Inclusivity in global research' questionnaire.** (PDF)

## Acknowledgments

We thank the participants for their time in providing data for this study, and the NIMHANS research team for their support in data collection and translation.

## Author contributions

**Conceptualization:** Mirja Koschorke, Graham Thornicroft, Petra C. Gronholm.

**Data curation:** Gurucharan Bhaskar Mendon.

**Formal analysis:** Sarah J. Parry, Gurucharan Bhaskar Mendon, Anish V. Cherian, Petra C. Gronholm.

**Funding acquisition:** Graham Thornicroft.

**Investigation:** Gurucharan Bhaskar Mendon, Adarsha Alur Manjappa.

**Methodology:** Sarah J. Parry, Gurucharan Bhaskar Mendon, Petra C. Gronholm.

**Project administration:** Adarsha Alur Manjappa.

**Supervision:** Anish V. Cherian, Petra C. Gronholm.

**Writing – original draft:** Sarah J. Parry.

**Writing – review & editing:** Sarah J. Parry, Gurucharan Bhaskar Mendon, Mirja Koschorke, Adarsha Alur Manjappa, Arya Thirumeni, Graham Thornicroft, Anish V. Cherian, Petra C. Gronholm.

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
