## [Decision Letter · Decision Letter 0]

3 Feb 2025

PONE-D-24-43039Mental health related stigma in a primary care setting in rural India: service user and carer perspectivesPLOS ONE

Dear Dr. Gronholm,

Thank you for submitting your manuscript to PLOS ONE. After careful consideration, we feel that it has merit but does not fully meet PLOS ONE’s publication criteria as it currently stands. Therefore, we invite you to submit a revised version of the manuscript that addresses the points raised during the review process.

**Along with the Reviewers' comments please add some more information on the included figure**

We look forward to receiving your revised manuscript.

Kind regards,

Eleni Petkari

Academic Editor

PLOS ONE

**Journal Requirements:**

GT, MK, and PCG were supported by the UK Medical Research Council (UKRI) for the Indigo Partnership (MR/R023697/1) award. GT has recently been supported by the National Institute for Health and Care Research (NIHR) Applied Research Collaboration South London (NIHR ARC South London) at King’s College Hospital NHS Foundation Trust. The views expressed are those of the author(s) and not necessarily those of the NIHR or the Department of Health and Social Care. For the purpose of open access, the author has applied a Creative Commons Attribution (CC BY) licence (where permitted by UKRI, ‘Open Government Licence’ or ‘Creative Commons Attribution No-derivatives (CC BY-ND) licence’ may be stated instead) to any Author Accepted Author Manuscript version arising from this submission.

AC received funding from an Intra mural grant from National Institute of Mental Health and Neuroscience, Bengalaru, India.

4. In the online submission form, you indicated that Data cannot be shared publicly because informed consent was not sought for sharing the data through other means than via request from the author. Data are available from authors via request. No reasonable request will be denied.

Reviewers' comments:

Reviewer's Responses to Questions

**Comments to the Author**

1. Is the manuscript technically sound, and do the data support the conclusions?

Reviewer #1: Partly

Reviewer #2: Yes

2. Has the statistical analysis been performed appropriately and rigorously? 

Reviewer #1: No

Reviewer #2: No

3. Have the authors made all data underlying the findings in their manuscript fully available?

Reviewer #1: No

Reviewer #2: No

4. Is the manuscript presented in an intelligible fashion and written in standard English?

Reviewer #1: Yes

Reviewer #2: Yes

5. Review Comments to the Author

**Reviewer #1:**  The topic is timely and interesting. However, the paper at this version lacks completeness and requires revion as suggested point by point in the document attached. All section of the paper required revision

**Reviewer #2:**  The subject of the study is quite interesting with the findings and the framework developed through the inductive process being impactful. However, there are still areas of improvement in the methodology, discussion and conclusion section.

Major focus is required on further literature review as the manuscript lacks global, regional and local background on the subject matter as well as compare and contrasts in discussion. Studies from LMICs, especially in South East Asia region, with similar study objectives requires to be reviewed and presented throughout the manuscript where reasonable.

Kindly address all the comments within the pdf file as well to refine the manuscript.

6. PLOS authors have the option to publish the peer review history of their article (what does this mean? ). If published, this will include your full peer review and any attached files.

**Do you want your identity to be public for this peer review?** For information about this choice, including consent withdrawal, please see our Privacy Policy .

Reviewer #1: **Yes: ** Hailemariam Mamo Hassen

Reviewer #2: No

---

## [Author Response · Author response to Decision Letter 1]

17 Jun 2025

Please see 'response to reviewers' provided as separate attached pdf file.

---

## [Decision Letter · Decision Letter 1]

15 Jul 2025

PONE-D-24-43039R1Mental health related stigma in a primary care setting in Karnataka, rural India: service user and carer perspectivesPLOS ONE

Dear Dr. Gronholm,

Thank you for submitting your manuscript to PLOS ONE. After careful consideration, we feel that it has merit but does not fully meet PLOS ONE’s publication criteria as it currently stands. Therefore, we invite you to submit a revised version of the manuscript that addresses the points raised during the review process.

Please check the Reviewers' suggestions for amending the Methods section by adding the information as required and for adapting the Discussion as suggested==============================

We look forward to receiving your revised manuscript.

Kind regards,

Eleni Petkari

Academic Editor

PLOS ONE

Journal Requirements:

Reviewers' comments:

Reviewer's Responses to Questions

**Comments to the Author**

1. If the authors have adequately addressed your comments raised in a previous round of review and you feel that this manuscript is now acceptable for publication, you may indicate that here to bypass the “Comments to the Author” section, enter your conflict of interest statement in the “Confidential to Editor” section, and submit your "Accept" recommendation.

Reviewer #1: All comments have been addressed

Reviewer #2: (No Response)

2. Is the manuscript technically sound, and do the data support the conclusions?

Reviewer #1: Yes

Reviewer #2: Yes

3. Has the statistical analysis been performed appropriately and rigorously? 

Reviewer #1: Yes

Reviewer #2: No

4. Have the authors made all data underlying the findings in their manuscript fully available?

Reviewer #1: Yes

Reviewer #2: Yes

5. Is the manuscript presented in an intelligible fashion and written in standard English?

Reviewer #1: Yes

Reviewer #2: Yes

6. Review Comments to the Author

Reviewer #1: Comments for the manuscript titled "Mental health related stigma in a primary care setting in Karnataka, rural India: service user and carer perspectives"

The authors have provided a "Response to previous round of review and revisions, which is a positive sign of engagement with feedback. However, a thorough review against PLOS ONE's criteria reveals several areas requiring improvement. While the study addresses a valid research question, clarity, methodological rigor in reporting, and comprehensive engagement with existing literature need to be strengthened.

Literature Review: The current version of the manuscript should be carefully re-evaluated to ensure the literature review comprehensively places the claims in context and fairly treats prior work.

Methods: The manuscript should clearly detail the specific qualitative methodology used (e.g., thematic analysis, grounded theory) and justify its selection. More extensive details on the interview guide, how interviews were conducted (e.g., duration, setting, language, who conducted them), and how data saturation was determined would enhance reproducibility. The data analysis process needs to be explained with greater rigor. This includes details on coding procedures, theme development, and how trustworthiness (e.g., credibility, transferability, dependability, confirmability) was ensured.

Ethics Statement: Ensure it clearly states how informed consent was obtained.

Data Availability: The manuscript needs to explicitly state where the data can be found, or if there are restrictions (e.g., due to participant privacy in qualitative data), these must be clearly specified and justified. For qualitative studies, this often involves depositing anonymized transcripts or a detailed data dictionary and codebook in a recognized repository.

Results

Support for Conclusions: Ensure the results directly answer the research questions posed in the introduction.

Discussion

Interpretation and Comparison: The discussion section should interpret the findings in light of previous research and their implications. The "Response to Reviewers" mentions that the discussion was revised to reflect the authors' interpretation of findings as related to the field. This is a good improvement, but the reviewer should critically assess if these comparisons are robust and if the discussion avoids being overly opinionated without sufficient comparative analysis.

Good Luck!!

Reviewer #2: The manuscript reads better after addressal of the comments from the reviewers. However, there are some major comment that needs to be highlighted:

1. It seems like the current study methodology is reviewing of the transcript available from previous multinational study (DOI: 10.1371/journal.pone.0258729). Therefore, the methods section may require rewriting as it is not a new study but using data and information from prior study. Consider secondary information review type of methodology highlighting the information availability through previous study.

2. Intercoder reliability or similar objective approaches would strengthen the rigor of the study rather than only speaking of agreeing upon on meetings (subjective approach).

3. The recommendations fall into conclusion section, not discussion section. Also, conclusion does not take any references.

7. PLOS authors have the option to publish the peer review history of their article (what does this mean? ). If published, this will include your full peer review and any attached files.

**Do you want your identity to be public for this peer review?** For information about this choice, including consent withdrawal, please see our Privacy Policy .

Reviewer #1: **Yes: ** Hailemariam Mamo Hassen(PhD)

Reviewer #2: No

---

## [Author Response · Author response to Decision Letter 2]

4 Aug 2025

The response for reviewers is attached as a pdf with the other manuscript documents.

---

## [Editor Report · Decision Letter 2]

8 Aug 2025

Mental health related stigma in a primary care setting in Karnataka, rural India: service user and carer perspectives

PONE-D-24-43039R2

Dear Dr. Gronholm,

We’re pleased to inform you that your manuscript has been judged scientifically suitable for publication and will be formally accepted for publication once it meets all outstanding technical requirements.

Kind regards,

Eleni Petkari

Academic Editor

PLOS ONE
---

## [Editor Report · Acceptance letter]

PONE-D-24-43039R2

PLOS ONE

Dear Dr. Gronholm,

I'm pleased to inform you that your manuscript has been deemed suitable for publication in PLOS ONE. Congratulations! Your manuscript is now being handed over to our production team.

Kind regards,

on behalf of

Dr. Eleni Petkari

Academic Editor

PLOS ONE